# Fixed Neural Network Steganography: Train the Images, not the Network

**Varsha Kishore,* Xiangyu Chen,* Yan Wang, Boyi Li & Kilian Weinberger**
Department of Computer Science
Cornell University
Ithaca, NY 14850, USA
`{vk352, xc429, yw763, bl728, kqw4}@cornell.edu`

## Abstract

Recent attempts at image steganography make use of advances in deep learning to train an encoder-decoder network pair to hide and retrieve secret messages in images. These methods are able to hide large amounts of data, but they also incur high decoding error rates (around 20%). In this paper, we propose a novel algorithm for steganography that takes advantage of the fact that neural networks are sensitive to tiny perturbations. Our method, Fixed Neural Network Steganography (FNNS), yields significantly lower error rates when compared to prior state-of-the-art methods and achieves 0% error reliably for hiding up to 3 bits per pixel (bpp) of secret information in images. FNNS also successfully evades existing statistical steganalysis systems and can be modified to evade neural steganalysis systems as well. Recovering every bit correctly, up to 3 bpp, enables novel applications that requires encryption. We introduce one specific use case for facilitating anonymized and safe image sharing. Our code is available at https://github.com/varshakishore/FNNS.

## 1 Introduction

Image steganography aims to hide a secret digital *message* within a *cover* image (Morkel et al., 2005) — ideally, through minimal alterations, such that only intended recipients are aware of the hidden secret. Steganography has been widely used in applications such as watermarking (Wolfgang & Delp, 1996), copyright certification (Lu, 2004) and private information storage (Srinivasan et al., 2004). Classic steganography tools use pixel statistics to hide information in images (Pevnỳ et al., 2010). Secret messages hidden with these methods can be recovered with 0% error, but to evade detection by steganalysis tools, they can only hide up to 0.5 bits per pixel (bpp) of information (Pevnỳ et al., 2010; Holub & Fridrich, 2012; Holub et al., 2014). Encouraged by data-driven deep learning techniques, recent methods propose training deep encoder-decoder networks to hide and recover up to 6 bpp of information in images (Zhang et al., 2019a; Zhu et al., 2018; Baluja, 2017; Wu et al., 2018; Hayes & Danezis, 2017). These methods do achieve higher bpp rates, but they also result in higher error rates for the retrieved messages (Zhang et al., 2019a).

Many applications have low error rate requirements for the steganography algorithm. In some scenarios, the hidden message has no redundancy for error correction, and there is zero tolerance for even a single incorrectly recovered bit. For example, if the secret message is encrypted, it will be a random bit string that must be recovered with zero error for successful decryption. In this paper, we propose a novel approach for image steganography that aims to simultaneously and reliably achieve high steganographic capacity and low error rates. Notably, we achieve **0.0% error** when encoding up to 3 bpp of information. We make **no assumptions** on the secret message and allow it to be any arbitrary bit string. We show that our method can be used with randomly initialized neural networks or in conjunction with pre-trained networks.

Unlike previous steganography methods that train deep networks to hide and recover messages in a specific dataset (Zhang et al., 2019a; Zhu et al., 2018; Baluja, 2017), our method is based upon

---

*Equal contribution.

a very different approach, which originated in the context of *adversarial attacks* on neural networks (Szegedy et al., 2013). Adversarial attacks are based on the key insight that deep neural networks are highly sensitive to small changes to the input. An adversary can therefore manipulate an image with imperceptible perturbations to influence the prediction of a neural network that uses this image as input. The last eight years have witnessed an outpouring of analysis (Kurakin et al., 2016b; Carlini & Wagner, 2016; Xu et al., 2017; Shafahi et al., 2018; Meng & Chen, 2017; Li & Li, 2017; Lu et al., 2017a; Graese et al., 2016; Dziugaite et al., 2016; Lu et al., 2017b; Gu & Rigazio, 2015; Kurakin et al., 2016a; Miyato et al., 2015; Nokland, 2015; Cisse et al., 2017; Hu et al., 2019; Guo et al., 2020; 2017) and methods (Liu et al., 2016; Moosavi-Dezfooli et al., 2016; Carlini & Wagner, 2017; Tramèr et al., 2017; Papernot et al., 2017; 2016; Biggio et al., 2013) to understand and create adversarial perturbations. Notably, it is fair to say that vulnerability to adversarial attacks is generally considered inevitable in most settings (Shafahi et al., 2018) and frustratingly hard to defend against (Carlini & Wagner, 2017; Dziugaite et al., 2016), especially when the target network architecture is known to the adversary (which is generally referred to as *white-box* setting).

Adversarial attacks are typically considered a nuisance, or a limitation of machine learning. However, in this paper we utilize their persistence and reliability as a desired feature for what we refer to as *Fixed Neural Network Steganography (FNNS)*. In a nutshell, FNNS is based on the following procedure (see Figure 1): We initialize a neural network (decoder) that takes as input an image and produces sufficient binary outputs. Given a secret message and a cover image, the sender (Alice) perturbs the original image in a fashion similar to adversarial perturbations (Madry et al., 2017). However, instead of targeting a single prediction bit (e.g. the classification of an image), the sender manipulates thousands or even millions of output bits simultaneously. The intended recipient (Bob) can use the same decoder network and recover the hidden message.

We show that FNNS reliably yields 0% error rate for hiding up to 3 bpp of information and lower error rates than current state-of-the-art methods for higher bit rates on multiple datasets. FNNS can also be used in conjunction with existing trained encoder-decoder methods (like SteganoGAN (Zhang et al., 2019a)) to further reduce the error rates obtained by the trained methods. Additionally, we show that FNNS evades existing statistical steganalysis methods (Boehm, 2014; Dumitrescu et al., 2002a;b; Böhme & Westfeld, 2004; Zhang & Ping, 2003) and can be made resistant to JPEG compression (Wallace, 1992) for low bpp. Finally, we introduce an example application of error-free steganography for anonymized image sharing: We replace faces in images with GAN generated substitutes that contain the original faces encrypted and hidden through FNNS — ensuring that only intended recipients (with the secret key) can recover the original images.

## 2 RELATED WORK

**Statistical image steganography** methods (Pevnỳ et al., 2010) typically pre-date the use of neural networks. Least-Significant Bit (LSB) methods modify lower-order bits of each pixel to encode a secret message (Van Schyndel et al., 1994; Wolfgang & Delp, 1996; Katzenbeisser & Petitcolas, 2000). Although compellingly simple and lossless, these methods are easily detectable (Dumitrescu et al., 2002a;b; Böhme & Westfeld, 2004; Zhang & Ping, 2003) and often lack robustness (Qin et al., 2010). Many statistical image steganography methods were proposed to evade detection by such steganalysis algorithms. Highly Undetectable Steganography (HUGO) (Pevnỳ et al., 2010), is one such method, that uses hand crafted features to measure distortion caused by modifying pixels and modifies pixels that cause the least amount of distortion. Wavelet Obtained Weights (WOW) (Holub & Fridrich, 2012) uses directional high-pass filters to find regions of the cover image with high texture and penalizes changes in low-textured regions. S-UNIWARD (Holub et al., 2014) is similar to WOW but is designed to work with non-spatial domains (e.g. the frequency domain). The main limitation of statistical methods is that the number of bits they encode is relatively low ($\leq 0.5$ bpp).

**Deep learning image steganography** methods have recently achieved impressive results in terms of bpp rates (Zhang et al., 2020b; Baluja, 2017; Rahim et al., 2018; Zhang et al., 2019b). In general, they mostly share a similar pipeline that can be trained end-to-end: An encoder network takes as input a cover image and a message that should be concealed within the image. From these inputs it generates a steganographic image that has hidden information but is visually similar to the cover image. A subsequent decoder network recovers the hidden message from the steganographic image. Multiple loss functions ensure that 1) the generated image is close to the original one; 2)

the decoder's output matches the secret message. Zhu et al. (2018)'s HiddenNet pioneered such an encoder-decoder pipeline, and their HiddenNet could hide up to 0.2 bpp with an error rate of $10^{-5}$. SteganoGAN (Zhang et al., 2019a) uses a slightly different encoder-decoder architecture and introduces an additional critic network that ensures the produced images look realistic, i.e. like a natural image. The authors show experiments of hiding up to 6 bpp of information with an error rate of 5-30% (depending on how many bits are hidden). In a similar vein, AdvSGAN (Li et al., 2021) hides up to 1 bpp by learning an image steganography scheme that plays an adversarial game between a restricted neural coder and a critic. Deep Steganography (Baluja, 2017), ISGAN (Dong et al., 2018), Attention Based Data Hiding (Yu, 2020), Universal Deep Hiding Zhang et al. (2020a) and End-to-end CNN for Image Steganography (Rahim et al., 2018) use similar encoder-decoder architectures to hide and recover structured images instead of random bits. These methods assume the secret message is an image, which allows them to learn image priors that aid in hiding the secret image. Invertible networks have also been explored to hide images within images (Jing et al., 2021; Lu et al., 2021). Despite handling a large number of bits, these end-to-end neural approaches also share a set of disadvantages: 1) the error rate for the recovered messages is very high, 2) they assume access to hundreds or even thousands of training images from the target domain to train encoder and decoder pairs, and 3) there is little recourse if the model produces an image with high error rate or distortions. There are also methods that have explored hiding messages in physical photographs (Wengrowski & Dana, 2019; Tancik et al., 2020), but our work focuses on digital images.

**Imperceptible image perturbations.** Adversarial examples (Goodfellow et al., 2014) are inputs to machine learning models that an attacker has intentionally designed to cause the model to make a mistake. Many approaches (Szegedy et al., 2013; Goodfellow et al., 2014; Guo et al., 2019; Xu et al., 2020; De Palma et al., 2021; Yuan et al., 2019) try to construct adversarial examples by perturbing image pixels. Moosavi-Dezfooli et al. (2017) propose a systematic algorithm for computing universal perturbations that many deep neural networks are highly vulnerable to. Su et al. (2019) propose a differential evolution method to generate low-dimensional one-pixel adversarial perturbations that change the output of a classification network. Athalye et al. (2018b) create adversarial examples that are robust to affine image transformations, noises, and other distortions. Projected gradient descent (PGD) (Madry et al., 2017) is one of the most widely used algorithms to generate adversarial examples by adding small perturbations to the input. This method iteratively updates the input with gradient descent until a desired output is obtained. The input is projected, or more precisely clipped to be within $[-\epsilon, \epsilon]$ at the end of every step. This precaution ensures that the perturbations stay reasonably small for all pixels and remain imperceptible. Ghamizi et al. (2019) propose performing stenography by finding perturbations using PGD with a classification network. However, the amount of information they are able to hide is low and they need multiple images to hide long messages.

## 3 FIXED NEURAL NETWORK STEGANOGRAPHY

**Setting.** Let $\boldsymbol{X} \in [0,1]^{H \times W \times 3}$ be an RGB color image with height $H$ and width $W$. Further, let $\boldsymbol{M} \in \{0,1\}^{H \times W \times D}$ be a message that we are trying to conceal in $\boldsymbol{X}$, where $D$ specifies the number of bits we need to hide per pixel.[1] We assume there are two parties involved – the **sender**, Alice, who hides $M$ in $\boldsymbol{X}$, creating $\tilde{\boldsymbol{X}}$; and the **receiver**, Bob, who extracts $\boldsymbol{M}$ out of $\tilde{\boldsymbol{X}}$. Given a decoder network $F : [0,1]^{H \times W \times 3} \rightarrow [0,1]^{H \times W \times D}$, our goal is to generate a perturbed image $\tilde{\boldsymbol{X}}$, which is close to $\boldsymbol{X}$ (according to metrics specified in the subsequent section) such that $F(\tilde{\boldsymbol{X}}) = \boldsymbol{M}$.

**Encoding and Decoding.** In order to generate image $\tilde{\boldsymbol{X}}$, we use an approach similar to adversarial perturbations (Madry et al., 2017). Given the cover image $\boldsymbol{X}$ and the ground truth message $\boldsymbol{M}$, the sender (Alice) solves the following optimization problem over the perturbed image $\tilde{\boldsymbol{X}}$:

$$\min_{\tilde{\boldsymbol{X}}} \underbrace{\langle \boldsymbol{M}, \log F(\tilde{\boldsymbol{X}}) \rangle + \langle (1 - \boldsymbol{M}), \log(1 - F(\tilde{\boldsymbol{X}})) \rangle}_{L_{BCE}},$$
$$\text{s.t.} \|\boldsymbol{X} - \tilde{\boldsymbol{X}}\|_\infty \leq \epsilon \text{ and } 0 \leq \tilde{\boldsymbol{X}} \leq 1, \tag{1}$$

where $\langle , \rangle$ denotes inner-product across all $H \times W \times D$ dimensions and the objective function is the binary cross entropy (BCE) loss. The first of the two linear constraints enforces that the maximum

---

[1]For convenience we assume the message is of length $H \times W \times D$. In practice, if the message length is not a multiple of $H \times W$ we can simply ignore the unused output bits during optimization.

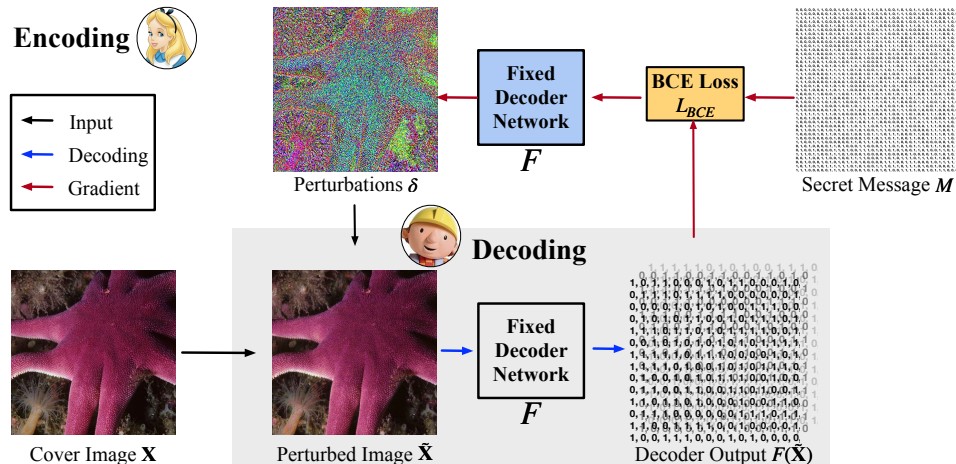

Figure 1: FNNS workflow: Alice (sender) encodes the message $M$ into image $\tilde{X}$ such that $F(\tilde{X}) = M$; Bob (receiver) decodes the message with the same decoder $F$. Alice generates perturbed image $\tilde{X}$ by using the gradient from the loss between the decoder output $F(\tilde{X})$ and secret message $M$.

---

**Algorithm 1** Adversarial Attack for Message hiding

---

1: Inputs: decoder network $F$, cover image $X$, secret message $M$
2: Hyper-parameters: learning rate $\alpha > 0$, perturbation bound $\epsilon > 0$, optimization steps $n > 0$, max L-BFGS iterations $k > 0$
3: $\tilde{X} \leftarrow X$
4: **for** $n$ iterations **do**
5:     $\tilde{X} = LBFGS(F(\tilde{X}), M, L_{\mathrm{BCE}}, k)$       ▷ Take $k$ steps to optimize $L_{\mathrm{BCE}}(F(\tilde{X}), M)$.
6:     $\delta \leftarrow clip_{-\epsilon}^{\epsilon}\{\tilde{X} - X\}$       ▷ Clip pixel value changes exceeding $\pm\epsilon$.
7:     $\tilde{X} \leftarrow clip_0^1\{X + \delta\}$       ▷ Clip pixel values to $[0, 1]$.
8: **return** $\tilde{X}$

---

perturbation does not exceed a value of $\epsilon$ and stays imperceptible. The second constraint enforces that the perturbed image, $\tilde{X}$, is a well-defined image, with pixel values ranging within $[0, 1]$.

Our optimization algorithm is outlined in Algorithm 1, where $clip_0^1(x) = \max(\min(x, 1), 0)$. We use the unconstrained L-BFGS (Fletcher, 2013) algorithm to optimize the objective with respect to $\tilde{X}$. We used L-BFGS because it keeps track of second order gradient statistics and results in faster optimization. To ensure that the constraints are not violated, we project the solution back into the feasible region after $k$ steps. We assume that the recipient (Bob) has access to the same network $F$ as the sender (Alice). He can then recover the concealed message $M$ by computing $F(\tilde{X})$.

**Decoder weights and initialization.** Our encoding procedure optimizes the image $\tilde{X}$ directly and considers the weights of $F$ fixed throughout. This gives us the freedom to explore multiple ways to initialize $\tilde{X}$ and to obtain weights for the decoder $F$, yielding three distinct variants of our method: 1) FNNS-R: $F$ is a *random* network and $\tilde{X}$ is initialized to be the cover image $X$. When a random network is used, the sender and the receiver only need to share the architecture of the decoder network and the random seed used to initialize its weights; the actual weights of the network do not have to be shared. Additionally, if the image quality of the perturbed image is low, a different random decoder can easily be initialized with a new random seed and the optimization can be repeated. 2) FNNS-DE: Given a trained encoder-decoder pair (from any of the prior neural work mentioned in section 2), we can define $F$ to be the *pre-trained* decoder and initialize $\tilde{X}$ as $Enc(X, M)$, where $Enc$ is the trained encoder that is paired with the decoder $F$. With this initialization, a part of the message $M$ is already encoded into $\tilde{X}$ such that $F(\tilde{X}) \approx M$. As a result, the optimization is much faster. However, the encoding step sometimes deteriorates image quality, and it is hard for the optimization algorithm to "recover" in terms of quality in such cases. 3) FNNS-D: $F$ is a *pre-trained* decoder and $\tilde{X}$ is initialized to be the cover image $X$. With a trained decoder, messages can be hidden in images using both perturbations and by training weights

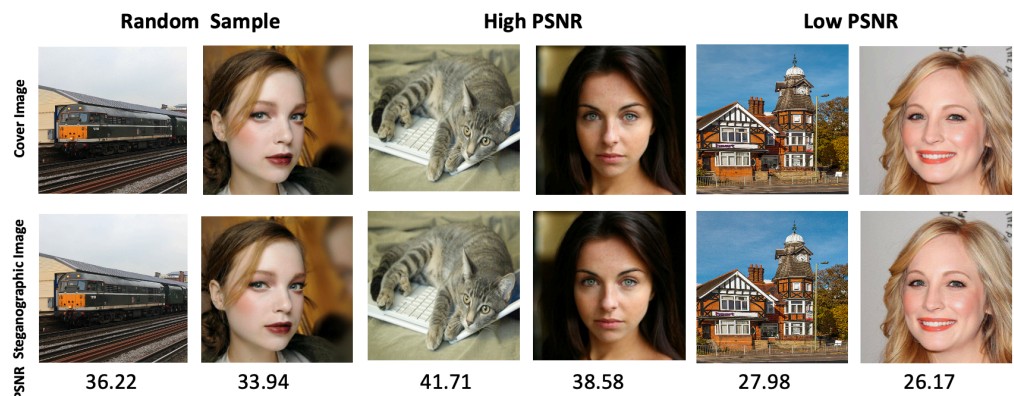

Figure 2: Examples of cover and steganographic image pairs from MS-COCO and CelebA. In each image FNNS-D is used to hide 4 bpp. (Zoom in for image details.)

conducive to hiding information in the images. As a result, the output image quality is better and the optimization does not suffer from getting stuck in bad local optima.

**Decoder architecture.** Prior work on adversarial attacks has explored network designs that are especially *robust* to small perturbations (Xu et al., 2020). In contrast, for our decoder we want network architectures that are particularly *susceptible* in that regard. We develop our method based on the *basic* decoder of SteganoGAN (Zhang et al., 2019a), a 4-layer convolutional neural network that takes an $H \times W \times 3$ RGB image as input and outputs a bit string $\{0,1\}^{H \times W \times D}$ (after rounding). We choose SteganoGAN because it has been shown to achieve state-of-the-art performance for hiding arbitrary messages in images. For our experiments, we set the pre-trained encoder and decoder for FNNS-D and FNNS-DE, to be the trained SteganoGAN encoder and decoder. For the FNNS-R random network we empirically explore many variations with different depths, widths, normalization layers, and activation functions, and evaluate them w.r.t. bit error rates, PSNR, and SSIM.[2] Due to space constraints, we summarize our results in Appendix B. Throughout, we use 128 hidden channels in FNNS-R and 32 in FNNS-D and FNNS-DE. This is because for trained SteganoGAN models, increasing the number of hidden units leads to no significant improvements in accuracy or image quality (see Appendix E), but slows down the optimization process.

### 3.1 EVALUATION METRICS

We use three popular metrics to evaluate our results. 1. The bit error rate, $\frac{\|M - \lfloor F(\tilde{X}) \rceil\|_1}{HWD}$, where $\lfloor \cdot \rceil$ denotes rounding function, measures how many bits are incorrectly recovered. 2. Peak signal-to-noise ratio (PSNR) is a common metric used to measure image distortions between $X$ and $\tilde{X}$ and has been shown to be correlated with human evaluation scores (Isola et al., 2017). 3. Structural Similarity Index (SSIM) is another metric used to measure the similarity between two images. SSIM differs

$$\mathbf{MSE} = \frac{1}{HW} \sum_{i=1}^{H} \sum_{i=1}^{W} \left[ \boldsymbol{X}_{i,j} - \tilde{\boldsymbol{X}}_{i,j} \right]^2$$

$$\mathbf{PSNR} = 20 \log_{10}(max_{\boldsymbol{X}}) - 10 \log_{10}(\mathbf{MSE})$$

$$\mathbf{SSIM} = \frac{(2\mu_{\boldsymbol{X}}\mu_{\tilde{\boldsymbol{X}}} + c_1)(2\sigma_{\boldsymbol{X}\tilde{\boldsymbol{X}}} + c_2)}{(\mu_{\boldsymbol{X}}^2 + \mu_{\tilde{\boldsymbol{X}}}^2 + c_1)(\sigma_{\boldsymbol{X}}^2 + \sigma_{\tilde{\boldsymbol{X}}}^2 + c_2)}$$

Table 1: Metrics to evaluate image quality. Note that $c_1, c_2$ are small stabilization constants.

from PSNR in that it tries to capture the change in structural information as opposed to considering pixel-wise changes. PSNR and SSIM between two images $X$ and $\tilde{X}$ with maximum possible pixel value $max_{\boldsymbol{X}}$, averages $\mu, \tilde{\mu}$, standard deviation $\sigma, \tilde{\sigma}$ and co-variance $\sigma_{\boldsymbol{X}\tilde{\boldsymbol{X}}}$ are defined in Table 1.

## 4 STEGANOGRAPHY RESULTS AND DISCUSSION

**Experimental Setup** We evaluate FNNS on three diverse datasets – a scenic image dataset Div2k (Agustsson & Timofte, 2017), a 2D object detection dataset MS-COCO (Lin et al., 2014) and a human face dataset CelebA (Liu et al., 2015). For each dataset, we use the provided test/validation images (if unavailable, we use the first 100 images in the dataset for validation). We randomly

---

[2]We also explored the use of fully connected networks but found them to yield far higher error rates, while being slower to be optimize and requiring larger amounts of memory.

| Dataset | Method | Error Rate (%) ↓ | | | | PSNR ↑ | | | | SSIM ↑ | | | |
|---------|--------|-------|--------|--------|--------|-------|--------|--------|--------|-------|--------|--------|--------|
| | | 1 bit | 2 bits | 3 bits | 4 bits | 1 bit | 2 bits | 3 bits | 4 bits | 1 bit | 2 bits | 3 bits | 4 bits |
| CelebA | SteganoGAN | 3.94 | 7.36 | 8.84 | 10.00 | 25.98 | 25.53 | 25.70 | 25.08 | 0.85 | 0.86 | 0.85 | 0.82 |
| | FNNS-R | 0.14 | 1.80 | 5.28 | 15.17 | 39.79 | 35.12 | 33.40 | 32.53 | 0.96 | 0.89 | 0.84 | 0.79 |
| | FNNS-D | **0.00** | **0.00** | **0.00** | 3.17 | 36.06 | 34.43 | 30.05 | 33.92 | 0.87 | 0.86 | 0.71 | 0.84 |
| | FNNS-DE | **0.00** | **0.00** | **0.00** | **2.58** | 21.16 | 20.85 | 20.67 | 21.03 | 0.71 | 0.68 | 0.63 | 0.64 |
| Div2k | SteganoGAN | 5.12 | 8.31 | 13.74 | 22.85 | 21.33 | 21.06 | 21.42 | 21.84 | 0.76 | 0.76 | 0.77 | 0.78 |
| | FNNS-R | 0.02 | 0.18 | 3.29 | 10.88 | 35.31 | 30.73 | 28.99 | 28.60 | 0.93 | 0.85 | 0.78 | 0.76 |
| | FNNS-D | **0.00** | **0.00** | 0.01 | 5.45 | 29.30 | 26.25 | 22.90 | 25.74 | 0.82 | 0.73 | 0.53 | 0.65 |
| | FNNS-DE | **0.00** | **0.00** | 0.01 | **1.75** | 18.54 | 18.02 | 17.16 | 17.38 | 0.60 | 0.53 | 0.39 | 0.60 |
| MS-COCO | SteganoGAN | 3.40 | 6.29 | 11.13 | 15.70 | 25.32 | 24.27 | 25.01 | 24.94 | 0.84 | 0.82 | 0.82 | 0.82 |
| | FNNS-R | 0.04 | 0.32 | 2.16 | 10.38 | 34.68 | 30.79 | 29.32 | 28.22 | 0.91 | 0.84 | 0.79 | 0.74 |
| | FNNS-D | **0.00** | **0.00** | **0.00** | 13.65 | 37.94 | 34.51 | 27.77 | 34.78 | 0.95 | 0.90 | 0.72 | 0.89 |
| | FNNS-DE | **0.00** | **0.00** | **0.00** | **1.74** | 22.52 | 22.35 | 21.02 | 21.33 | 0.77 | 0.74 | 0.62 | 0.64 |

Table 2: Performance of FNNS and its variants with different bpp rates. All values shown are averaged over 100 images. We bold the lowest error rate numbers up to statistical significance. Note that the 0.00 numbers are exactly zero.

initialized the message bit strings; each bit in the string is independently sampled from a Bernoulli-distribution with probability $\frac{1}{2}$. [3] The hyper-parameters used for Algorithm 1 are as follows: perturbation bound $\epsilon = 0.3$, optimization steps $n = 100$, and L-BFGS iterations $k = 10$ with early stopping if the output has zero error. [4] In cases where the image quality of $\tilde{X}$ is poor, we restart optimization with a different learning rate $\alpha$. Concretely, we set the learning rate to 0.1 and change it to 0.05 or 0.5 if the output image gets a PSNR lower than 20. We train SteganoGAN models for only one epoch for FNNS-D and FNNS-DE, as we observe that a fully-trained (32 epochs) SteganoGAN decoder over-fits to its training objective such that it's hard to use it for FNNS. Appendix G shows the result of using SteganoGAN models trained for 32 epochs.

**Quantitative Comparison.** We compare FNNS with SteganoGAN, the current state-of-the-art method, in Table 2. [5] In addition (not shown in the table), we also compare with steganography methods that hide lower payload messages (that is < 0.5 bpp messages): HUGO (Pevnỳ et al., 2010), UNIWARD Holub et al. (2014), WOW (Holub & Fridrich, 2012) and HiDDeN (Zhu et al., 2018). These methods can achieve an error rate of 0% but only for < 0.5 bpp messages. In contrast, FNNS also achieves 0% error for significantly higher bit rates as shown in Table 2. FNNS-D achieves the best performance in terms of both low error rates and good image quality. FNNS-DE is also able to achieve error rates of 0%, but the corresponding image quality is worse when compared to FNNS-D. From Table 2, we also see that for Div2k it is hard to achieve 0% error with 3 bpp. Div2k is a small dataset with only 800 images and as a result there is not enough diversity to train a flexible SteganoGAN model; this is evidenced by the fact that 0% error for Div2k with 3 bpp can be achieved by using a model trained on MS-COCO.

**Qualitative Comparison.** Several qualitative examples are presented in Figure 2 (more images from all the method variants can be found in Appendix A). Even with 4 bpp information hidden, the cover and stenographic images look identical. There are a few examples where the stenographic images look pixelated, especially when a random network is used. This is either because the image is over-optimized to get a low bit error rate or because of a bad random seed. To produce a better stenographic image, the random network can be re-initialized (if using a random network), the optimization can be stopped earlier or the hyper-parameters of FNNS can be changed slightly (in order to change the optimization problem).

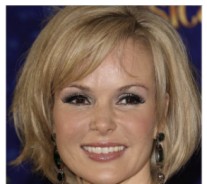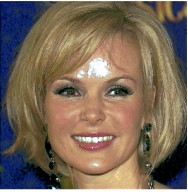

Figure 3: An Example of artifacts that arise when using an out-of-domain model. Left: original image; right: a steganographic image generated from SteganoGAN.

**Domain independence.** The performance of trained encoder-decoder models like SteganoGAN degrade for out-of-domain cover images. For example, a SteganoGAN (Zhang et al., 2019a) model trained on Div2k images (which contains primarily landscapes) produces noticeable artefacts when

---

[3]We did not observe that any of the randomly initialized strings are harder. But, it was harder to optimize an all 0s or all 1s message with some decoder initializations.

[4]We continue optimizing for 25 steps after achieving 0 error in order to ensure that even after converting $\tilde{X}$ into integer color values (integers between 0 to 255), $F(\tilde{X}) = M$.

[5]Results with higher bpp are shown in Appendix F.

tested on facial images from CelebA (Figure 3). The only recourse to correcting artifacts or quality would be to train another model with in-domain data. In contrast, FNNS-R is completely domain independent. Although FNNS-D uses a trained decoder, we find it to be just as robust against domain shift. We explain these findings by the fact that the secret message is hidden through an optimization procedure that is very robust to the exact filter values of the decoder network.

**Optimization time.** Most encoder-decoder deep stenography methods require many hours of training on hundreds of images. However, once trained, it takes less than a second for inference with the model given any cover image and message. FNNS-D and FNNS-DE perform a per-image optimization process for encoding a message in an image and this takes on average 10 seconds for 1-2bpp and 20 seconds for 3-4bpp. For FNNS-R encoding a message takes about 3 minutes because optimizing the random network is a harder task. Appendix C has a table showing the amount of time required to use each method variant with different bpp rates.

**JPEG Compression.** A desirable property for any steganography system is robustness against lossy image compression. Unfortunately the goal of imperceptibility is inherently at odds with most image compression methods, which by design aim to remove imperceptible, and therefore unnecessary, information. JPEG (Wallace, 1992) is a lossy compression method for digital images that transforms an image into frequency space and removes high frequency components via quantization and

| bpp | Error Rate (%) | PSNR | SSIM |
|-----|----------------|-------|------|
| 0.1 | 0.06 | 22.9 | 0.49 |
| 0.2 | 0.34 | 22.43 | 0.48 |
| 0.5 | 5.85 | 22.18 | 0.46 |
| 1.0 | 32.03 | 22.65 | 0.48 |

Table 3: Performance of JPEG-resistant FNNS on MS-COCO. FNNS achieves low error rates when the bpp rate is low.

rounding. Past work has shown that adversarial perturbations can be removed via JPEG compression (Dziugaite et al., 2016) and trained encoder-decoder steganography networks also fail to be resistant to JPEG compression. Modifications to the loss functions can improve resistance to compression, but significantly increase error rates. We can improve the robustness of FNNS to JPEG compression by adding a JPEG layer (with quality factor 80) in our optimization pipeline, in which the back-propagated gradients are approximated with identity transformation (Athalye et al., 2018a). We evaluate on MS-COCO and show results in Table 3. Although we can successfully encode 0.1 bpp (enforcing BCE loss on 10% of pixels) at only 0.06% error, it is fair to say that bit-rates beyond 0.5 bpp are currently still out of reach. Alternative approaches that we may explore in the future include adopting the Shadow Attack approach from (Ghiasi et al., 2020) to find larger semantically meaningful perturbations and incorporating global smoothness constraints.

## 5 STEGANALYSIS

Steganalysis tools are used to identify whether an image has a hidden message. Broadly, these tools can be divided into two categories – statistical and neural steganalysis. The former uses statistical methods to detect steganography, while the latter trains a neural network to distinguish between natural images and steganographic images.

| Method | 1 bpp | 2 bpp | 3 bpp | 4 bpp |
|--------|-------|-------|-------|-------|
| FNNS | 8 | 18 | 22 | 31 |
| FNNS-D | 17 | 13 | 15 | 8 |
| FNNS-DE | 2 | 3 | 0 | 0 |

Table 4: Detection Rates (in %) obtained by using StegExpose on CelebA steganographic images. Lower is better.

Statistical steganalysis methods were developed to detect LSB (least significant bit) steganography (Gupta et al., 2012) in lossless images. Boehm (2014) presents StegExpose, a library combining many existing steganalysis techniques including Sample Pairs (Dumitrescu et al., 2002b), RS Analysis (Fridrich et al., 2001), Chi Squared Attack (Westfeld & Pfitzmann, 1999), and Primary Sets (Dumitrescu et al., 2002a). We show the detection rate of StegExpose for the Div2K dataset in in Table 4. As seen in the table, some images are detected, but the detection rates are low in comparison with traditional methods that are detected with an accuracy of over 75% (Zhu et al., 2018). It is worth noting that even if the image is detected as stegnographic, messages cannot be accessed if they are encrypted.

Neural steganalysis makes use of recent advances in deep learning to detect images with hidden information (Ye et al., 2017; You et al., 2020; Lerch-Hostalot, 2021). Neural networks are powerful and are capable of accurately detecting steganographic images from most existing methods, even when they hide less than 0.5 bpp of information. Recent work has explored using signal from steganalysis systems in the encoding process to evade detection (Tang et al., 2019; Bernard et al.,

2021; Zhang et al., 2019b). Along similar lines, we use the gradient signal from a differentiable steganalysis network, SiaStegNet(You et al., 2020), and include its classification score as an auxiliary detection loss in FNNS optimization, and find perturbations such that SiaStegNet does not classify the image as steganographic. We evaluate this method's feasibility and compare results in Table 5. The detection rates are shown in the right most section of the table. Since SiaStegNet performs random cropping when loading the image, it is hard to guarantee 100% detection avoidance. Still, with < 2 bpp, we see that FNNS achieves 0.0% error rate, while 80% of generated images can successfully evade detection. Furthermore, the weight for this detection loss can be adjusted to obtain higher quality images with a slightly higher error rate (see Appendix H). We also present the detection results for hiding < 1 bpp of information in Appendix H. In future, we would like to investigate to what extent we can evade detection by black box steganalysis networks and find robust defenses against them.

| Dataset | Error Rate (%) ↓ | | | | PSNR ↑ | | | | Detection Rate (%) ↓ | | | |
|---|---|---|---|---|---|---|---|---|---|---|---|---|
| | 1 bit | 2 bits | 3 bits | 4 bits | 1 bit | 2 bits | 3 bits | 4 bits | 1 bit | 2 bits | 3 bits | 4 bits |
| CelebA | 0.00 | 0.00 | 0.00 | 0.00 | 21.04 | 20.60 | 20.18 | 19.70 | 13 | 30 | 58 | 100 |
| Div2k | 0.00 | 0.00 | 0.00 | 0.02 | 20.32 | 18.07 | 17.66 | 18.23 | 3 | 20 | 100 | 100 |
| MS-COCO | 0.00 | 0.00 | 0.00 | 0.01 | 22.28 | 22.35 | 21.30 | 21.36 | 7 | 12 | 14 | 83 |

Table 5: Performance of FNNS-D with auxiliary detection loss from SiaStegNet (You et al., 2020) trained on MS-COCO. The detection rate is measuring the number of steganographic images that can be detected as steganographic by SiaStegNet.

# 6  APPLICATION - FACE ANONYMIZATION

As shown in table 2, we can reliably obtain 0% error rates for hiding 3 bpp messages. The ability to recover the bit string with zero error allows us to encrypt the message, which is currently the only provable way to safeguard private information. This property enables us to create a protocol where an encrypted message or image is sent from one party to another, and the file is itself an indistinct looking image — allowing users to use photo sharing webpages or social media as a medium of transmission. Below, we describe how FNNS can be used to create a protocol to share images that are anonymized to the public but not to trusted recipients.

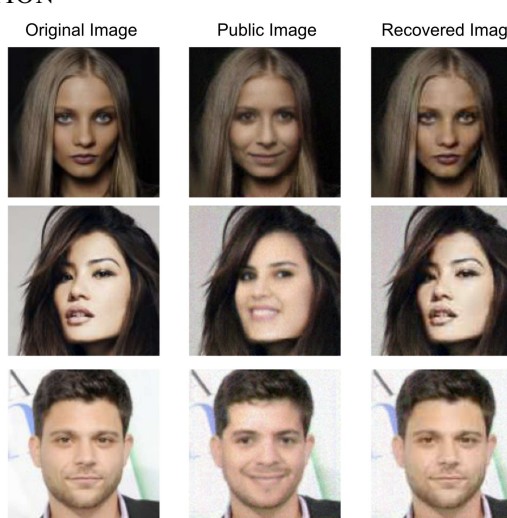

Figure 4: The left column contains original images. The middle column shows the result of replacing the face with a fake face, which contains a encrypted form of the original face steganographically hidden through FNNS. The right column shows the image recovered by FNNS decoding the encrypted face, decrypting and re-inserting it.

**Motivation.**  Social media platforms have become a central part of our communication, with photo sharing as one of the center activities (e.g. Instagram, Facebook). There are however well documented dangers with such practices. If photos become publicly available, the owner loses control. Snapshots intended to amuse friends or grandparents can "go viral", with traumatic consequences for individuals portrait in the photos. Further, public photos are quickly indexed by image search engines (e.g. https://clearview.ai/) and can haunt individuals decades later. We use FNNS to facilitate a mechanism that allows users to share pictures with their friends on social media, but if these are leaked (e.g. through wrong privacy settings on social media pages, or careless behavior), the identity of the people in the images is preserved.

**FNNS face anonymization.**  Current methods for face anonymization either significantly alter the image by using modifications (like blurring or masking) (Neustaedter et al., 2006; Boyle et al., 2000),

Original Image          Public Image          Recovered Image

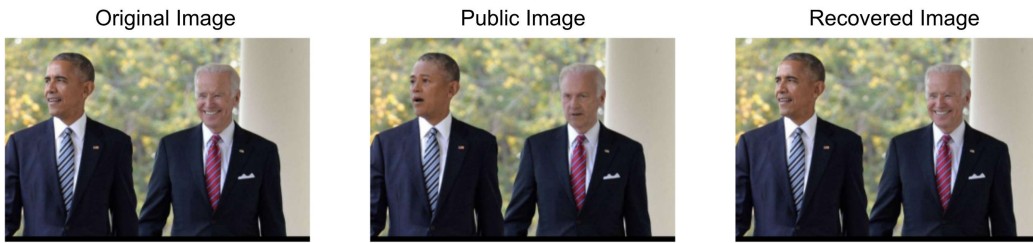

Figure 5: Example of anonymizing multiple faces in an image. In this example each private face is hidden within the corresponding fake face that replaces it.

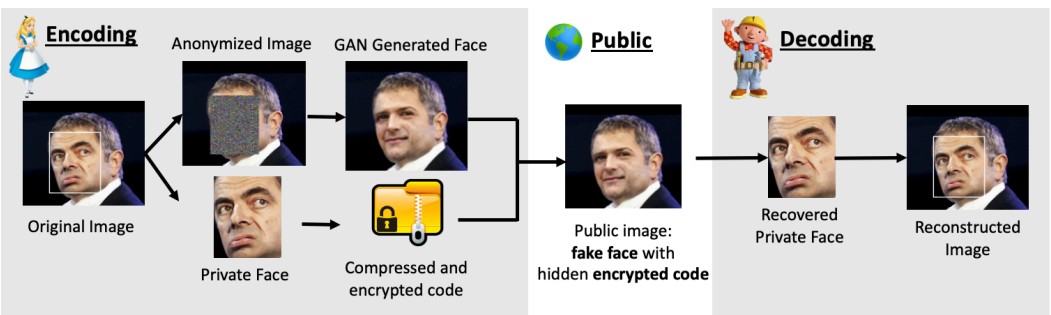

Figure 6: Pipeline for FNNS face anonymization.

or remove the face completely and replace it with a "fake" face (Hukkelås et al., 2019). The former approach can still leak information and cause the quality of the image to degrade, and the latter approach non-reversibly changes the image. We propose a novel approach for face anonymization using FNNS (see Figure 4).

We make use of DeepPrivacy (Hukkelås et al., 2019) to detect faces in the **original image** (left) and replace them with plausible looking GAN-generated faces to obtain an anonymized **public image** (middle image). Because the original face is cut before a fake face is generated, the public image leaks no information about the sensitive face and can safely be shared. Since FNNS can reconstruct the message perfectly, each original private face can be hidden inside its corresponding fake face in the public image. To ensure the private face is secure, we first compress it (e.g. using JPEG (Wallace, 1992)) and then encrypt it into a cryptographically secure bit string (e.g. using AES (Standard, 2001)). The intended recipient can decode this bit-string, using the decoder network and decrypt the image with a private key to obtain the **recovered image** (right image). Any third party will only observe the public image without any access to private information. See Figure 6 for a layout of the application pipeline. Because all information is hidden inside the faces in the public image, it is naturally compatible with existing image sharing pages, and the encryption / decryption portion can be realized easily in practice e.g. through a simple browser plugin. As each fake face contains all the information required to reconstruct the original private face, this approach can be applied to an arbitrary number of (non-overlapping) faces in an image. An example is presented in Figure 5.

## 7 CONCLUSION

In this paper, we propose a novel and effective algorithm for image steganography based on techniques from the adversarial attacks literature. We show that our method can achieve very low error rates while making visually unnoticeable changes to the input image. In contrast to prior work, we are able to achieve error rates of exactly 0% for up to 3 bpp, which in turn enables new applications. In the future we plan to explore other variants and extension of FNNS. One useful variant would be to explore new spaces in which to conduct the optimization. Currently we use RGB space, but other color spaces like YCbCr or non-spatial spaces like the frequency space can be used as well with FNNS. Using different spaces could result in better image quality and could allow for easier encoding of constraints. For instance, to find perturbations robust to JPEG compression, FNNS can be used in only the low frequency region. In the future, we would also like to explore meta-learning algorithms for finding networks that are conducive to FNNS optimization.

ACKNOWLEDGEMENTS

This research is supported by grants from the National Science Foundation NSF (IIS-2107161, IIS-1724282), the DARPA Techniques for Machine Vision Disruption grant (HR00112090091), the Cornell Center for Materials Research with funding from the NSF MRSEC program (DMR-1719875), and SAP America. We would like to thank Oliver Richardson, Katie Luo, Kate Donahue and our reviewers for their valuable feedback and insightful comments.

ETHICS STATEMENT

Steganography is a tool and it's usage depends on the user. Steganography can potentially be used for activities with negative societal impact, such as hidden communication for criminal activities, or espionage. But it can also be used for beneficial applications like preventing copyright infringements through watermarking, tracing illegal images on social media or anonymizing publicly shared images (as shown in the paper).

REPRODUCIBILITY STATEMENT

For our experiments, we use three publicly available datasets as mentioned in section 4. We use common metrics to evaluate our results and these metrics are explained in subsection 3.1. Our method is outlined in Algorithm 1 and we have also clearly described all the hyper-parameter values we used to obtain our results in section 4. Our code is available at Our code is available at https://github.com/varshakishore/FNNS.

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

# A  ADDITIONAL QUALITATIVE EXAMPLES

## A.1  EXAMPLES FROM FNNS-R

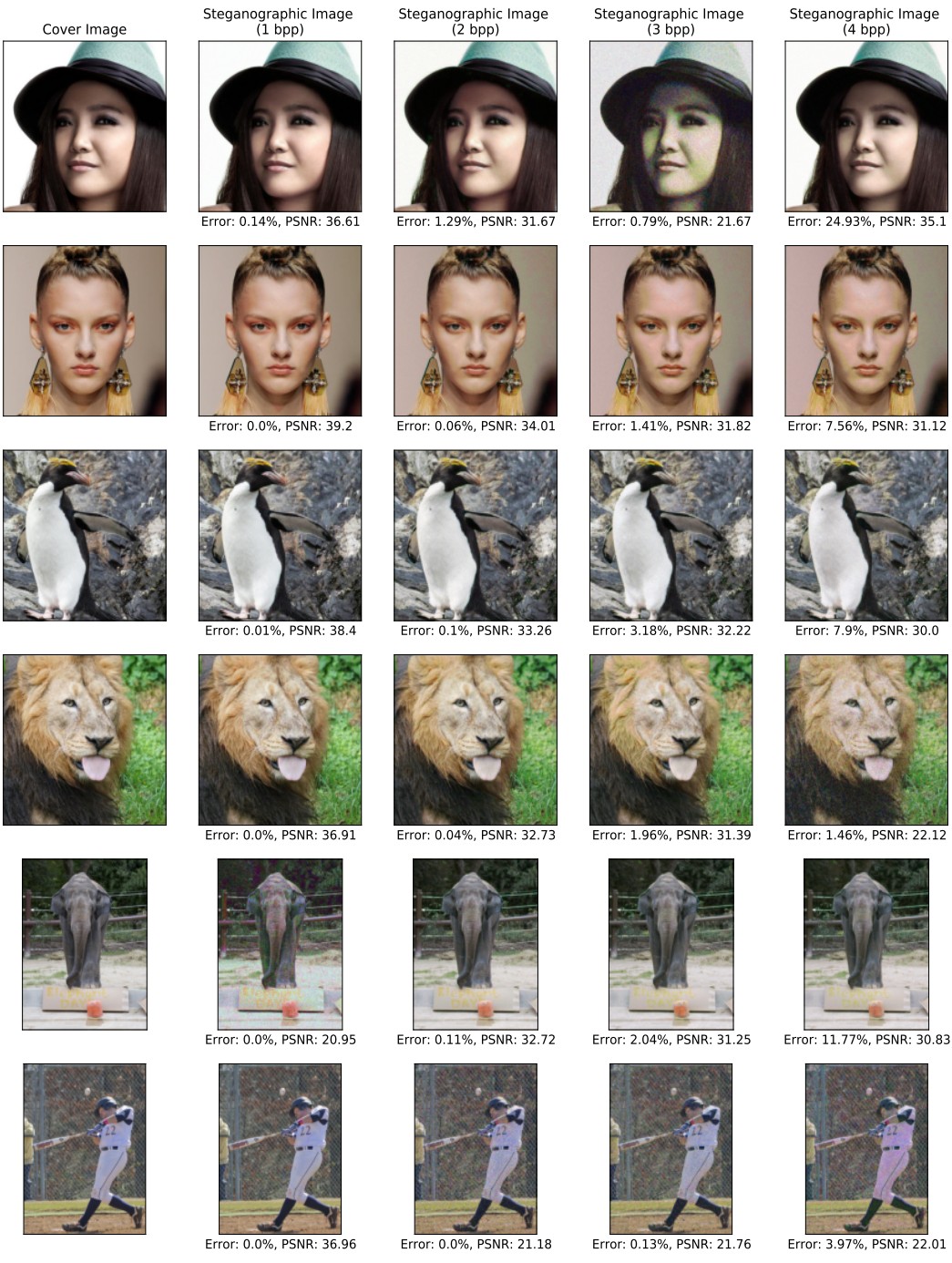

Figure 7: Examples of images with different amounts of hidden information. The first two images are from CelebA, the next two are from Div2k and the last two are from MS-COCO. PSNR values vary mostly due to the randomness of the networks. For images with low PSNR values the quality improves with a re-initialized network.

## A.2 EXAMPLES FROM FNNS-D

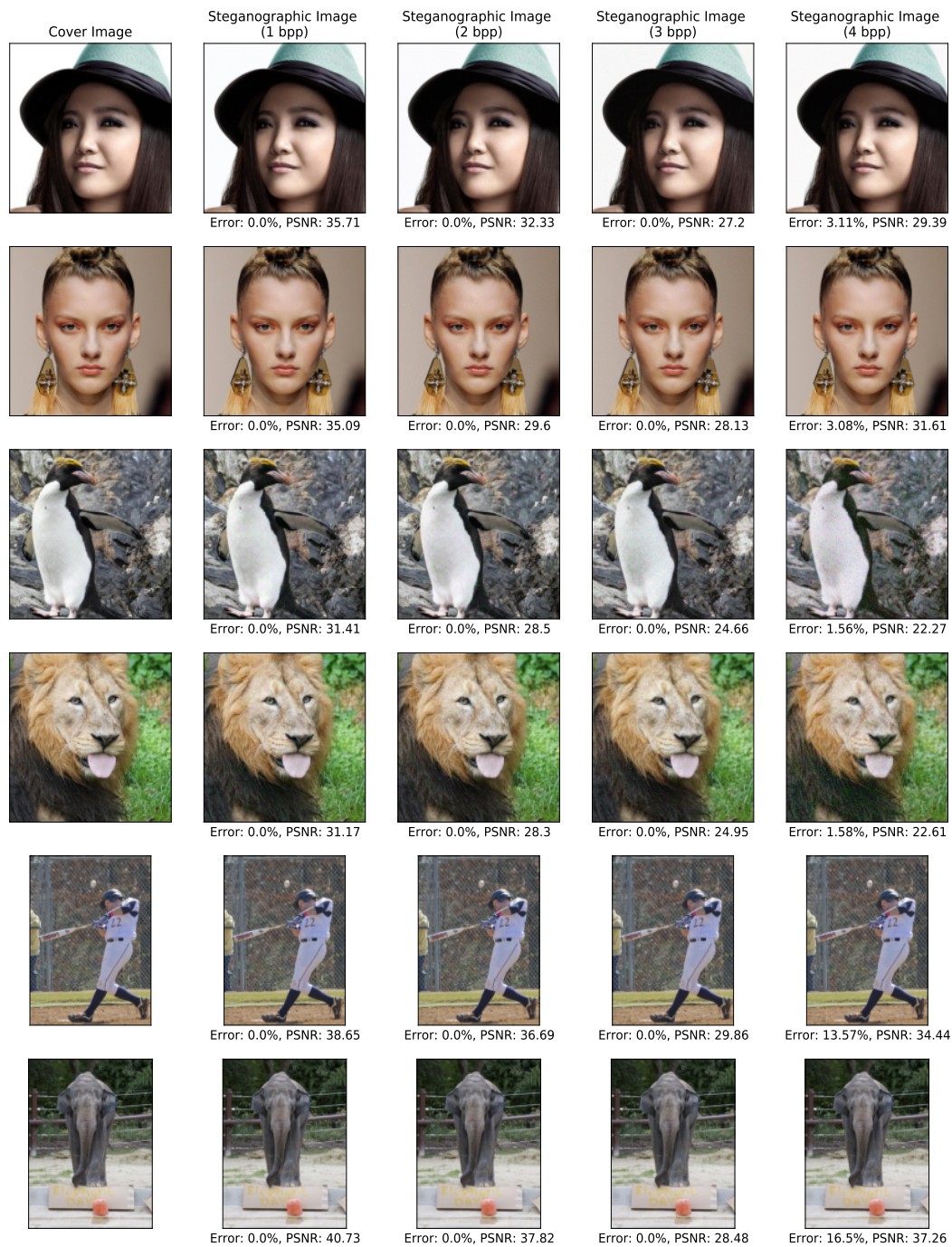

Figure 8: Examples of images with different amounts of hidden information. The first two images are from CelebA, the next two are from Div2k and the last two are from MS-COCO.

## A.3 EXAMPLES FROM FNNS-DE

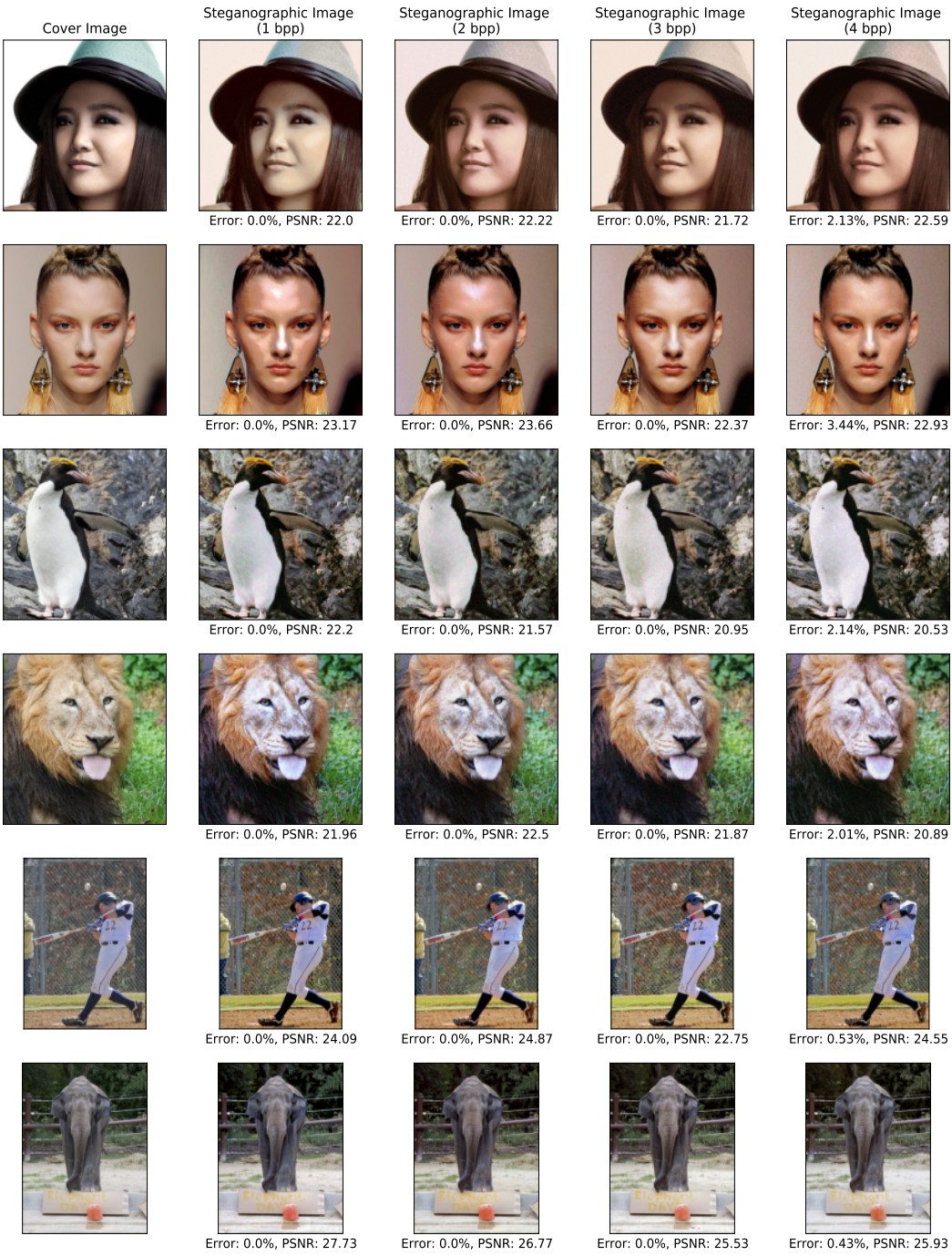

Figure 9: Examples of images with different amounts of hidden information. The first two images are from CelebA, the next two are from Div2k and the last two are from MS-COCO. As seen in row 1 or row 4, the color of the image changes sometimes. This is because the trained Steganogan network learns that information can be hidden by changing the color and maintaining the structure.

# B    RANDOM DECODER ARCHITECTURE

The decoder introduced in section 3 can be any network that takes an image as input and produces sufficiently many binary outputs, but some architectures are better suited for the task at hand. Unlike most prior work, that explores networks that are robust to adversarial attack (Xu et al., 2020), we find ourselves looking for architectures that are susceptible to it. Consequently, in our search for a suitable decoder, we empirically explore many network architectures with different network depth, width, normalization and activation functions and evaluate them w.r.t. pixel error rates, PSNR, and SSIM. For the experiments in this sections, we test different architectures using 100 images from the training set of the Div2K (Agustsson & Timofte, 2017) dataset, with dimensions $512 \times 512 \times 3$ unless otherwise noted. Since, our input consists of images, we focus on convolutional networks (CNN) (Goodfellow et al., 2016).[6]

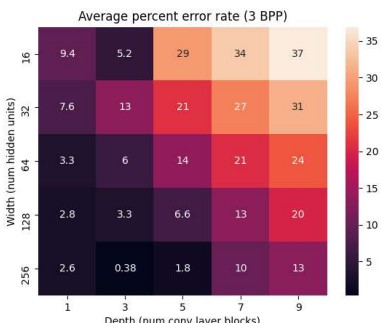

Figure 10: Error rates (%) of 3 bits based on various widths and depths. We calculate the error rate on average over 100 images from Div2k. The lower the better.

**Network depth and width.**    Figure 10 presents results of convolutional networks with varying height and width, and their respective error rates. Similar to the finding in Wu et al. (2020), we observe that wider networks are less robust and that increasing the width can lead to networks that are more sensitive to perturbations. However, we notice that the opposite is true for depth; the deeper the network, the less sensitive it is to perturbations. Although a width of 256 seems clearly superior, it is fair to say that the optimization is also slower. Finding a perturbed version of a $512 \times 512 \times 3$ image with 3 bpp takes 1 minute for network with a width of 32, but 6 minutes with width 256 (on a NVIDIA GTX 1080 GPU).

**Normalization.**    As the weights of the decoder network are set randomly, the gradients can have high variance. In Table 6 we experiment with two types of normalization layers that do not require mini-batch statistics. We include Positional

| Normalization | Error Rate(%) ↓ | PSNR(dB)↑ | SSIM↑ |
|---|---|---|---|
| None | $36.1 \pm 9.5$ | 15.8 | 0.35 |
| PONO | $15.4 \pm 8.6$ | 17.7 | 0.34 |
| IN | $2.5 \pm 2.7$ | 23.9 | 0.55 |

Table 6: Results with different normalizations. We calculate the average across 10 images.

Normalization (PONO) (Li et al., 2019) and Instance normalization (IN) (Ulyanov et al., 2016) in our model. The results show that both normalization methods improve the error rates and both visual metrics, but IN performs the best in this setting. We hypothesize this is because normalization stabilizes the gradients and brings activations near the activation tipping points, which eases the optimization process. Unlike IN, PONO removes the structural statistics (Li et al., 2019), which might be informative to create small perturbations.

**Activation.**    One important design choice for neural networks are the activation functions. In Table 7 shows results for various popular choices. We observe that LeakyReLu results in better performance when compared to other activations like sigmoid or tanh. The PSNR and SSIM values for sigmoid or tanh are (trivially) high because the optimization fails and very few changes are made

| Activation | Error rate (%)↓ | PSNR(dB)↑ | SSIM↑ |
|---|---|---|---|
| None | $1.8 \pm 3.9$ | 20.9 | 0.39 |
| ReLU | $7.1 \pm 9.6$ | 19.3 | 0.29 |
| LeakyReLU | $2.9 \pm 4.3$ | 24.7 | 0.57 |
| Sigmoid | $25.3 \pm 4.1$ | 39.5 | 0.96 |
| Tanh | $39.4 \pm 1.6$ | 45.5 | 0.95 |

Table 7: Error rates of different activation functions. We conceal 3 bits for each pixel. The model without any activation function performs best.

---

[6]We have also explored the use of fully connected networks but found them to yield far higher error rates, while being slower to optimize for.

to the input image. Omitting the activation layer, despite having low error rates, produces images of inferior quality. Adding the ReLU non-linearity makes the optimization easier and more stable.

The different architectures have different trade-offs in terms of memory, time, and performance. Wider and deeper networks require more time and memory. For our random network experiments, we settled on four convolutional layers (three intermediate layers and one output layer) with 128 hidden units each, along with instance normalization and ReLU activation.

## C  TIMING

For all our reported results, we used images with have a resolution of $512 \times 512$ (this which results in 1,048,576 hidden bits at 4-BPP). Table 8 shows the amount of time required to encode a message with different FNNS variants and different bit rates (with standard deviations in parentheses) on a NVIDIA GTX 1080 GPU. We also explored using images of different sizes and found that the encoding time scales approximately linearly with the number of pixels; for every factor of 4 increase in the number of pixels, the encoding time increases by a little less than a factor of 4. Note that the encoding optimization is significantly slower if run on a CPU. We have currently not optimized the code explicitly for CPU or mobile GPU cards, and this is an interesting avenue for future work.

| Dataset | Method | Time in seconds | | | |
| --- | --- | --- | --- | --- | --- |
| | | 1 bit | 2 bits | 3 bits | 4 bits |
| CelebA | FNNS-D | 9.77 (3.46) | 13.42 (5.17) | 30.81 (21.20) | 47.39 (15.32) |
| | FNNS-DE | 7.47 (2.20) | 11.31 (4.90) | 35.65 (12.32) | 45.39 (16.78) |
| | FNNS | 45.94 (5.23) | 124.85 (11.29) | 151.94 (7.42) | 152.18 (8.15) |
| Div2K | FNNS-D | 4.95 (1.08) | 10.53 (9.9) | 44.39 (5.27) | 44.29 (5.40) |
| | FNNS-DE | 4.86 (0.54) | 8.09 (5.05) | 43.82 (4.75) | 44.13 (6.73) |
| | FNNS | 42.18 (4.17) | 114.44 (6.48) | 156.91 (3.88) | 159.19 (4.63) |
| MS-COCO | FNNS-D | 7.41 (2.50) | 10.61 (5.9) | 37.72 (15.27) | 48.39 (9.40) |
| | FNNS-DE | 5.13 (1.97) | 7.04 (2.34) | 32.76 (16.20) | 48.29 (9.63) |
| | FNNS | 47.35 (4.23) | 131.85 (12.31) | 182.47 (6.49) | 184.39 (5.74) |

Table 8: Time in seconds for different methods

## D  UNCONSTRAINED OPTIMIZATION

As we saw in Equation 1, we have two constraints- 1) to ensure that the pixels are between 0 and 1 and 2) to ensure that no pixel changes by more than $\epsilon$. We tried translating this optimization problem defined in Equation 1 to an unconstrained optimization problem by reparameterizing to check if unconstrained optimization yielded better results. Constraint 1 can easily be relaxed by reparameterizing $\boldsymbol{X} \in [0, 1]^{H \times W \times 3}$ to $\boldsymbol{Z} \in \mathcal{R}^{H \times W \times 3}$ by applying an inverse sigmoid transform $\boldsymbol{Z} = \sigma^{-1}(\boldsymbol{X})$ and optimizing $\boldsymbol{Z}$ instead of $\boldsymbol{Z}$. We can also relax the second constraint with a slightly more involved process by computing the softmax over the set of admissible pixel values. However, we see no improvements by using these relaxations. We believe that the main source of error is not the projection step (that is clipping), but rather the restrictions (subpixels remain in [0,1] and change by at most $\delta$) on the perturbation. Regardless of the parameterization, these restrictions remain and in some cases will not be satisfiable.

## E  STEGANOGAN HIDDEN LAYER SIZE

Table 9 shows difference in performance when using 32 hidden channels vs 128 hidden channels. As seen in the table the difference is quite small. However, using networks with a smaller hidden size speeds up both the training of the network and using it for FNNS. Hence, for FNNS we use networks with 32 hidden channels.

| Method | error rate (%) | | | | PSNR | | | | SSIM | | | |
|---|---|---|---|---|---|---|---|---|---|---|---|---|
| | 1 bit | 2 bits | 3 bits | 4 bits | 1 bit | 2 bits | 3 bits | 4 bits | 1 bit | 2 bits | 3 bits | 4 bits |
| SteganoGAN - 32 hidden units | 5.12 | 8.31 | 15.74 | 22.85 | 23.53 | 23.66 | 22.86 | 22.87 | 0.83 | 0.82 | 0.8 | 0.82 |
| SteganoGAN - 128 hidden units | 5.97 | 10.28 | 18.11 | 24.93 | 20.98 | 21.52 | 21.54 | 21.75 | 0.76 | 0.76 | 0.77 | 0.78 |

Table 9: Performance of SteganoGAN with 32 hidden units and 128 hidden units.

## F  HIGHER BITS PER PIXEL

Table 10 shows the performance of FNNS with large message payloads (5-6 bpp). SteganoGAN results in a very high error rate. FNNS-based methods can achieve lower error rates with its optimization. However, in some cases the image quality is not great. Since the SteganoGAN model cannot encode and decode 5-6 bpp well, FNNS works better with randomly initialized weights than with the pre-trained decoder.

| Dataset | Method | Error Rate (%) ↓ | | PSNR ↑ | | SSIM ↑ | |
|---|---|---|---|---|---|---|---|
| | | 5 bit | 6 bits | 5 bits | 6 bits | 5 bit | 6 bits |
| Celeba | SteganoGAN | 32.15 | 31.16 | 19.51 | 21.82 | 0.74 | 0.79 |
| | FNNS-R | 14.14 | 16.27 | 18.68 | 17.86 | 0.18 | 0.16 |
| | FNNS-D | 15.3 | 18.41 | 12.94 | 12.99 | 0.07 | 0.07 |
| | FNNS-DE | 15.95 | 17.88 | 18.22 | 16.56 | 0.17 | 0.13 |
| Div2k | SteganoGAN | 31.44 | 35.35 | 20.05 | 20.34 | 0.79 | 0.8 |
| | FNNS-R | 13.01 | 15.98 | 16.71 | 16.46 | 0.25 | 0.26 |
| | FNNS-D | 18.12 | 19.67 | 12.34 | 12.3 | 0.13 | 0.14 |
| | FNNS-DE | 16.03 | 17.63 | 14.74 | 14.94 | 0.2 | 0.21 |

Table 10: Performance of FNNS and its variants with 5-6 bpp. All values shown are averaged over 100 images.

## G  OPTIMIZE FULLY TRAINED STEGANOGAN

For FNNS-D and FNNS-DE we used SteganoGAN models trained for 1 epoch as mentioned in section 4. This ensured that the model was flexible enough to achieve 0% error. Table 11 shows the result of using SteganoGAN models trained to completion for 32 epochs. As seen in the table, the error rate decreases significantly for FNNS-D when compared with the numbers in Table 2. The learning rate was set to 0.5 and the number of optimization steps was set to 200 when using a SteganoGAN model trained for 32 epochs. All other hyper-parameters were the same as listed in section 4.

| Dataset | Method | Error Rate (%) ↓ | | | | PSNR ↑ | | | | SSIM ↑ | | | |
|---|---|---|---|---|---|---|---|---|---|---|---|---|---|
| | | 1 bit | 2 bits | 3 bits | 4 bits | 1 bit | 2 bits | 3 bits | 4 bits | 1 bit | 2 bits | 3 bits | 4 bits |
| mscoco | SteganoGAN | 2.42 | 4.02 | 7.76 | 10.56 | 27.49 | 27.04 | 26.64 | 26.73 | 0.88 | 0.87 | 0.84 | 0.85 |
| | FNNS-D | 0.00* | 0.01 | 1.31 | 2.88 | 27.30 | 26.98 | 26.56 | 26.61 | 0.86 | 0.86 | 0.84 | 0.84 |
| celeba | SteganoGAN | 3.94 | 7.36 | 8.84 | 10.00 | 25.98 | 25.53 | 25.70 | 25.08 | 0.85 | 0.86 | 0.85 | 0.82 |
| | FNNS-D | 1.00 | 0.88 | 2.73 | 3.38 | 26.02 | 25.54 | 25.77 | 25.08 | 0.84 | 0.85 | 0.85 | 0.81 |
| Div2k | SteganoGAN | 5.12 | 8.31 | 13.74 | 22.85 | 21.33 | 21.06 | 21.42 | 21.84 | 0.76 | 0.76 | 0.77 | 0.78 |
| | FNNS-D | 0.00* | 1.12 | 4.37 | 11.96 | 21.01 | 20.98 | 20.88 | 21.27 | 0.71 | 0.74 | 0.67 | 0.68 |

Table 11: Performance obtained when using a SteganoGAN trained for 32 epochs. In the table, * implies that the value is 0 rounded to two decimal places but it is not exactly 0.

## H  STEGANALYSIS EVASION

In section 5, we see that we can add signal from SiaStegNet into the FNNS optimization process by adding an auxiliary SiaStegNet loss term, to evade detection from SiaStegNet. In Table 12 we follow the same setting and show results for hiding $< 1$ bpp of information. To hide less than 1 bpp of information we only compute $L_{BCE}$ with a subset of pixels in the image. We see that FNNS is able to achieve $0.0\%$ error rate and low detection rates for hiding $< 1$ bpp of information.

Table 5 shows that the image quality is a little low when the additional SiaStegNet loss term is added. To increase the image quality, we can decrease the weight on the SiaStegNet loss term (from 1000 to 100) and the results of doing so are shown in Table 13. This results in a slight increase in the error rate but the error rate is still under 1% for <3 bpp.

| Dataset | Error Rate (%) ↓ | | | PSNR ↑ | | | Detection Rate (%) ↓ | | |
|---------|---------|---------|---------|---------|---------|---------|---------|---------|---------|
| | 0.1 bit | 0.2 bit | 0.5 bit | 0.1 bit | 0.2 bit | 0.5 bit | 0.1 bit | 0.2 bit | 0.5 bit |
| CelebA | 0.00 | 0.00 | 0.00 | 18.04 | 18.05 | 20.72 | 0 | 0 | 0 |
| Div2k | 0.00 | 0.00 | 0.00 | 25.02 | 24.95 | 25.13 | 24 | 17 | 18 |
| MS-COCO | 0.00 | 0.00 | 0.00 | 21.55 | 21.67 | 21.27 | 15 | 8 | 8 |

Table 12: Performance of FNNS-D with the auxiliary detection loss from SiaStegNet (You et al., 2020) for hiding < 1 bpp of information.

| Dataset | Error Rate (%) ↓ | | | | PSNR ↑ | | | | Detection Rate (%) ↓ | | | |
|---------|-------|--------|--------|--------|-------|--------|--------|--------|-------|--------|--------|--------|
| | 1 bit | 2 bits | 3 bits | 4 bits | 1 bit | 2 bits | 3 bits | 4 bits | 1 bit | 2 bits | 3 bits | 4 bits |
| CelebA | 0.00 | 0.00 | 0.55 | 1.35 | 26.42 | 19.28 | 18.66 | 25.07 | 24 | 52 | 66 | 93 |
| Div2k | 0.00 | 0.02 | 0.21 | 3.95 | 26.15 | 19.12 | 19.23 | 21.14 | 37 | 69 | 84 | 99 |
| MS-COCO | 0.02 | 0.01 | 0.01 | 13.69 | 36.27 | 30.29 | 18.37 | 34.87 | 2 | 19 | 49 | 55 |

Table 13: Performance of FNNS-D with a lower weight on auxiliary detection loss from SiaSteg-Net (You et al., 2020) trained on MS-COCO.

