# OpenReview forum: "Fixed Neural Network Steganography: Train the images, not the network"
_ICLR.cc/2022/Conference — ICLR 2022 Poster_

### Official Review · Reviewer_vPjm · 2021-10-25

**Correctness:** 3
**Technical Novelty And Significance:** 3
**Empirical Novelty And Significance:** 3
**Recommendation:** 8
**Confidence:** 4

**Main Review:**

# Strong Points
* FNNS constitute a novel approach in the field of steganography leveraging adversarial attack techniques to hide a secret message in an image.
* The paper is well written and easy to follow
* The presented application of face anonymization is interesting.
* The authors present that FNNS is capable of evading steganalysis frameworks.

# Weak Points
* I am wondering about the computation time of the proposed approach? It appears that FNNS has to process each image individually with L-BFGS, which might be relatively time-intensive. On the other hand, previous techniques can hide secret information in a matter of a single forward pass. This might make FNNS inefficient at deployment in real-world scenarios.
* I am wondering about the motivation of the authors to choose the L-BFGS algorithm? In the adversarial attack community many other, maybe better or faster adversarial attack algorithms had been proposed. Did the authors experiment with other attack algorithms such as PGD, DeepFool, or C&W?
* The paper misses related works. Just to mention a few: [1,2,3,4,5,6,7,8], which should be discussed and ideally differentiated and compared to.
* The source code is not provided? Will the code be publicly available?

[1] Light field messaging with deep photographic steganography; CVPR 2019
[2] Stegastamp: Invisible hyperlinks in physical photographs; CVPR 2020
[3] UDH: Universal Deep Hiding for Steganography, Watermarking, and Light Field Messaging - NeurIPS 2020
[4] HiNet: Deep Image Hiding by Invertible Network; ICCV 2021
[5] Large-Capacity Image Steganography Based on Invertible Neural Networks; CVPR 2021
[6] Deep Multi-Image Steganography with Private Keys; 2021
[7] Enhancing the Security of Deep Learning Steganography via Adversarial Examples; Mathematic 2020

# Additional Questions
* The motivation for the naming of FNNS is not entirely clear to me. What does the **Fixed** in FNNS stand for? Aren’t all other neural network-based steganography techniques using fixed neural networks?


**Summary Of The Paper:**

This work proposes Fixed Neural Network Steganography (FNNS), an algorithm for steganography leveraging adversarial attacks to hide a secret message in an image. The authors propose three variants of FNNS, show its performance under steganalysis, and propose face anonymization as a possible use case of FNNS.

**Summary Of The Review:**

In my opinion, FNNS can provide a contribution to the steganography community. The proposed algorithm is novel and includes sufficient results and the evaluation and analysis appear valid. The authors also show an interesting use case of FNNS. However, the paper still has some weak points the authors should address to increase the quality and completeness of the paper.

====== Edit November 23 ======
Raised my score to Accept after reading the author's response.

---

> ### Author Response · Authors · 2021-11-17
> **Response to reviewer vPjm**
>
> Thank you for the thoughtful comments. We provide clarification to your questions and concerns below:
>
> * Computational time: This is a valid concern. However, depending on the variant that is used, the added time might not be a lot. If FNNS-DE is used it only takes about 5 seconds to optimize an image and this process can probably still be optimized a lot more. For applications that require 0% error, trained encoder-decoder methods cannot be used even though they are fast because they currently cannot achieve 0% error rates. Our method is also especially useful for use cases when you don't have a lot of data to train a model or are worried about failures due to domain shift. As shown in figure 3, trained models are vulnerable to domain shift.
> * Optimizer and attack algorithm: We used L-BFGS because it keeps track of second order statistics and is a good single image optimization algorithm. We did try PGD and found that it was much slower. It also produced images of worse quality because using PGD resulted in a larger number of tiny changes that caused the image to be pixelated. C&W would likely be even slower than PGD and we therefore did not try it. We can potentially use DeepFool by changing the loss function in our optimization. We will definitely explore these variants for the final version, and report whether it leads to performance improvements.
> * Missing citations: Thank you for pointing these out. We will add these citations into the paper and discuss how they compare to our proposed method. In general, these citations refer to papers that are hiding images within images as opposed to hiding an arbitrary bit string within an image. Some of them are also specifically designed to work with non-digital images.
> * Source code: The code will be released with the camera-ready version. It will include both the scripts to reproduce our results and general purpose code for image steganography.
> * Motivation for title: Other neural network steganography algorithms train encoder-decoder style networks and update the weights of the network during training. In contrast, we keep the weights of our network fixed throughout the optimization and only make changes to the image.
>
> Please let us know if you have any additional questions or concerns.

---

> > ### Comment · Reviewer_vPjm · 2021-11-23
> > **Thank you for your response.**
> >
> > I thank the authors for their response. Overall the authors addressed my concerns and I urge the authors to include the discussion regarding computational time and the choice of the L-BFGS method in their final submission. I believe that this work is ready for publication and hence raise my score to accept.

---

### Official Review · Reviewer_Cqai · 2021-10-28

**Correctness:** 3
**Technical Novelty And Significance:** 3
**Empirical Novelty And Significance:** 2
**Recommendation:** 5
**Confidence:** 5

**Main Review:**

There has been a form of misunderstanding in some recent publications in the ML community that claim to do steganography. Stricto sensu, these papers do hide information in a cover content and this secret message is typically not visible to the human eye. Yet the problem is that those techniques are easily detectable by SOTA opponents (steganalysis algorithms) and thus one does not reach the required level of secrecy in this multimedia security context. Also, in practice, it is unthinkable to use an algorithm that cannot embed an encrypted message which means that the message must be recovered with 0% error all the time. This is why the steganography community has been working on the problem of minimizing detectability given a message length to embed and does not allow to trade message retrieval exactness with detectability.
Fortunately, the present paper seems to be aware of the problem and attempts to remediate this issue by achieving 0% error using ML-based message embedding.

There are a number of claims in the introduction with which I do not agree :
- Steganography is presented as a superset of other problems such as watermarking or copyright certification. I argue that these are separate problems with separate solutions even if they borrow some ideas from one another. This is again about the misunderstanding mentioned above.
- Classic steganography techniques « can only hide up to 0.4 bits per pixel  » in order to avoid detection. First, such an empirical result is highly dependent on the cover image. There are some for which one can embed much more and other possibly less. So this must be an average bpp. More importantly, I do no think any steganographic scheme can claim to achieve 0% detection with 0.4 bpp. The usual experimental setup is to provide a curve showing the detection rate as a function of bpp payload. This curve will be dependent on the image dataset and the chosen opponent (usually a SOTA steganalyzer).
- StegoGAN is not SOTA steganography.. it might be ML-based SOTA steganography. But the mainstream steganography does better than this notably regarding detectability.

Concerning related works and novelty, the paper is not the first to exploit adversarial image to improve steganography see for example:

[1] W. Tang, B. Li, S. Tan, M. Barni, and J. Huang. 2019. CNN-Based Adversarial Embedding for Image Steganography. IEEE Transactions on Information Forensics andSecurity14,8(2019),2074–2087

[2] Bernard, S., Bas, P., Pevný, T., & Klein, J. (2021, June). Optimizing Additive Approximations of Non-Additive Distortion Functions. In Proceedings of the 2021 ACM Workshop on Information Hiding and Multimedia Security (pp. 105-112).


The authors use clipping to prevent the stego and cover to have too high pixelwise differences and to prevent out-of-range pixel value assignments.. In practice, most image formats use 1 byte for each pixel (or 3 bytes for RGB). So it would have been more convincing to clip image between 0 and 255 and then clip modifications to integers values.

The quality of a steganographer should not be evaluated through reconstruction errors such as PSNR, MSE or SSIM because they do not say much about detectability. This is what Table 5 reveals. The detection rates are too high to entrust the algorithm for achieving secret communication. BTW, detection rate is a bit ambiguous term, it is true positive rate ? It would be better to show classifier error rate.
The security of the steganography must be evaluated through SOTA steganalyzer error rates. The authors use ref. [You et al. 2020] as SOTA steganalyzer. Others are:

[3] Boroumand, M., Chen, M., & Fridrich, J. (2018). Deep residual network for steganalysis of digital images. IEEE Transactions on Information Forensics and Security, 14(5), 1181-1193.

[4] Xu, G. (2017, June). Deep convolutional neural network to detect J-UNIWARD. In Proceedings of the 5th ACM Workshop on Information Hiding and Multimedia Security (pp. 67-73).


The authors say on page 6, that they include some comparison with HUGO but I could not find it in the mentioned table (maybe I missed something). As a baseline, I think the authors should at least compare themselves to UNIWARD [Holub et al. 2014] although there are now more secure steganography schemes in the mainstream steganography literature.

Besides, one of the main weakness of the presented study is that the achieved zero error for message retrieval is only empirical while with statistical image steganography zero error is a built-in property. So in practice, one has to trust the generalization performances of the model in order to send an encrypted message.

While section 6 is somewhat interesting, I think this use case is quite narrow. The security relies on 0% error + encryption but many other concurrent approaches can do that relying on the cover produced by the face GAN. So why the proposed approach should be used ? It is because the message to embed is large ? I agree that, here, detectability is not the top priority. But I am unsure about the interest of publicly sharing someone’s picture with the face of someone else.  If two parties want to share pictures they can use private messaging apps with encryption.

**Summary Of The Paper:**

This submission deals with steganographic paradigm : hiding a secret message in a cover content while remaining undetectable. The authors use a purely ML-based pipeline. They intend to achieve zero message-retrieval-error (which is normally a must-have for steganographic approaches). They use a neural network decoder and exploit adversarial perturbations to compute the stego version of the cover image. The loss from which gradients originate is the sum of binary cross-entropies between decoder output and message to embed.


**Summary Of The Review:**

pros :
- it is clever to rely on adversarial examples and iterating to obtain a stego image

cons :
- the evaluation in terms of undetectability (which is what matters most in steganography) is insufficient: one should provide the error rate of a few SOTA opponents. The method must also be compared to gold standards steganographers.
- the zero error of message retrieval is only empirically checked and one cannot hope for perfect generalization performances

---

> ### Author Response · Authors · 2021-11-17
> **Response to reviewer Cqai**
>
> Thank you for the in-depth review. Here are the responses to your questions and concerns:
>
> * Phrasing: As per your suggestion, we will definitely clarify some of our wording in the final version. Specifically, we agree that application specific techniques can be used for watermarking or copyright certification, but steganography can also be used for these applications. We will clarify this and add citations. The statement saying traditional steganography methods can hiding up to 0.4bpp was an upper bound taken from prior literature. We will specify that this is for general purpose steganography (that is, no assumptions are made about the image or the message).
> * State-of-the-Art Methods: Do you have a particular method in mind that we should compare against? We’ve looked at the traditional methods (UNIWARD, HUGO, etc.) and there are neural steganalysis systems that detect steganography with these methods with high accuracy (about 75% or higher). There might be methods that make assumptions on the cover image or the message that is being encoded in order to achieve higher payloads and evade detection, but we did not find a general purpose method with a higher payload. If you could post them in this thread, we’d be happy to include them as additional comparisons if possible.
> * Clipping Pixels: What you describe, is essentially what we did. Only, our images are scaled to be between 0 and 1 and so we clip that steganographic image to also be in this range. Finally, we optimize for extra iterations to ensure that rescaling to 0..255 and quantizing to uint8 does not affect the decoding accuracy.  Please refer to footnote 3 in page 6.
> * Steganalysis: As you mentioned, recent work has shown that it is very hard to make steganography algorithms that are completely resistant to neural steganalysis even for low payloads. We show the detection rates for different bpp in table 5. We also show that the loss from a neural steganalyzer can be incorporated into our optimization to evade detection. Ultimately, the acceptable detection rate will always depend on the application (detection is not a priority for the application presented in section 6) and also dictate how many bpp one can hide.
> * Baselines: We compare with traditional methods, including HUGO, in the “Quantitative Comparison” section. We will add the numbers from additional traditional methods in this section but they deal with much smaller payloads.
> * Empirical performance: While theoretically possible that there are images for which one cannot obtain 0% error, we have never encountered a single such case at 2bpp across the thousands of images and across three independent data sets that we experimented with. It is important to emphasize that only the perturbations are generated by a neural network, but the actual steganography is obtained through an optimization process for which there is no notion of model generalization. Hence, a failure would mean that there is no feasible solution for an image and a particular network initialization. Finally, it might be important to keep in mind that the person who hides the message can immediately test if all bits are correctly hidden, and if not, can re-optimize with random restarts (e.g. after adding low variance noise or relaxing the image quality constraints). A future line of work we are interested in is proving theoretical guarantees about convergence with different initializations.
> * Application: GANs can be used to generate fake faces and anonymize the image. However, they replace the original face and it can no longer be retrieved. Our method allows us to design a mechanism by which a trusted receiver can recreate the original image.
> A private messaging app would require all the parties to move to using that app. In contrast, our proposed method can be implemented as a browser plugin that can be used in conjunction with _existing_ photo sharing methods. In many settings it can be much easier to convince your friends and family to install a small browser plugin, than to switch to a completely different social media platform.
> * Prior Work: Both the papers you mentioned (CNN-Based Adversarial Embedding for Image Steganography and Optimizing Additive Approximations of Non-Additive Distortion Functions) use existing embedding cost based steganography methods and propose mechanisms to add costs that help minimize detectability of specific steganalyzers. The methods are adversarial because they are trying to evade steganalysis. We instead use imperceptible adversarial perturbations directly to perform steganography with a neural decoder. The contribution of the mentioned papers is complimentary: similar to this prior work we can also add a loss term in our optimization to evade detection from steganalysis.
>
> (The last bullet point was added in an edit.)
>
> Please let us know if any have any additional questions or concerns.

---

> > ### Comment · Reviewer_Cqai · 2021-11-22
> > **Response to the authors' comments**
> >
> > I thank the authors for their answers. I am sorry to reply on the last day, I therefore do not expect authors to have time to iterate.
> >
> > I just wanted to clarify that :
> > - the novelty/position of the submission wrt papers I mentioned is clarified in my opinion
> > - the prior approaches that one must compare to are (typically) :
> >     x [3,4] for steganalysis. One must train these networks using cover/stego pairs where the stego ones are obtained with the authors approach. It is important to prove that if Eve plays after you, she cannot detect hidden messages.
> >     x Uniward for steganography. While other more recent papers can outperform Uniward, it remains a gold standard.

---

> > > ### Author Response · Authors · 2021-11-23
> > > **Response to reviewer Cqai**
> > >
> > > Thank you for the follow-up. Here are a few additional details:
> > >
> > > * For steganography, Uniward can hide up to 0.5 bpp and we will add this number to the quantitative comparison section of the paper.
> > > * For Steganalysis, we agree with the reviewer that for some applications, steganographic images being undetectable (when Eve plays after you) is important. However, even existing state-of-the-art steganography methods are already detectable with high accuracy for 0.5bpp (UNIWARD is 92.95% detectable and WOW is 93.28% detectable [1]). We used SiaStegNet [2] to test our steganography method with 0.5 bpp on MSCOCO and were able to detect the steganographic images with a comparable accuracy of 93%. We believe that if we add either an mse regularization term or a total variation regularization term in the FNNS optimization, the detectability will further reduce because smoother perturbations will be found. We will add a discussion section about this in the appendix. Note that SiaStegNet is shown to achieve comparable performance to SRNet. Since we only had a day, we trained SiaStegNet because it is smaller and has fewer parameters; we expect to see similar results with SRNet and will include results from both SiaStegNet and SRNet for 0.1-0.5bpp in the appendix. Additionally, we show in section 5 of the paper that if one were aware of the steganalysis system that was going to be used, the loss from that steganalyisis system can be added to FNNS to evade detection.
> > >
> > > [1] Boroumand, M., Chen, M., & Fridrich, J. (2018). Deep residual network for steganalysis of digital images. IEEE Transactions on Information Forensics and Security, 14(5), 1181-1193.
> > >
> > > [2] You, W., Zhang, H., & Zhao, X. (2020). A Siamese CNN for image steganalysis. IEEE Transactions on Information Forensics and Security, 16, 291-306.

---

### Official Review · Reviewer_NkWr · 2021-10-29

**Correctness:** 4
**Technical Novelty And Significance:** 3
**Empirical Novelty And Significance:** 3
**Recommendation:** 8
**Confidence:** 3

**Main Review:**

Strengths:
1. The paper is written clearly and easy to follow.
2. The observations regarding the limitations of an encoder - decoder architecture is novel and provides new insights to the field.
3. The empirical performance of the method is strong. The out-of-domain example is convincing and the decode accuracy of the method is very high. Moreover, the paper deals with the setting of an arbitrary binary message.

Weakness:
1. The parts regarding JPEG compression can be expanded. For example,
(a) what are the JPEG compression quality factors used for the experiment?
(b) It would be interesting to combine the approach with JPEG-resistant adversarial attacks. But it's understandable if this is outside the current scope of the paper.

**Summary Of The Paper:**

This paper proposes a method for image steganography based on adversarial attacks. Unlike previous works such as SteganoGAN, the proposed method fixes the decoder and performs an inference-time attack to produce a perturbation which maps to the correct message. Empirical results show the proposed method out-performs existing encoder decoder based design, and is also more robust in out-of-domain applications.

**Summary Of The Review:**

Overall I enjoyed reading this work. See main review for details.

---

> ### Author Response · Authors · 2021-11-17
> **Response to reviewer NkWr**
>
> Thank you for your supportive comments, we are glad you enjoyed the paper! The JPEG compression quality we used is 80 (we will emphasize this in the final version). We also tried with a quality factor of 75 (which is the PIL image processing library’s default JPEG compression factor) and obtained similar results.  Investigating JPEG-resistant adversarial attack is an interesting line of future research. We are especially excited about investigating semantic adversarial attacks because semantic attacks allow for changes of larger magnitude and these larger magnitude changes can potentially be more robust to JPEG compression. We will see ...

---

### Official Review · Reviewer_3seg · 2021-11-02

**Correctness:** 3
**Technical Novelty And Significance:** 2
**Empirical Novelty And Significance:** 3
**Recommendation:** 8
**Confidence:** 3

**Main Review:**

## Strengths
The observation that neural networks are sensitive to adversarial perturbations can be regarded as a feature and not a bug is very nice.
The method is simple and quite general. The experiments show that the run-time of the algorithm, while appropriate for high-throughput applications,
is reasonable (on the magnitude of a few minutes).
The experiments are relevant and the methodology seemed reasonable with a few minor issues.

The proposed application of hiding images is also very interesting. In my view, its usefulness depends on how easily the optimization procedure can be performed
on e.g. mobile devices.

On the whole, the paper is easy to follow and is generally well-written with only a few mistakes.

## Weaknesses
My main concerns are regarding some of the experimental procedures and results and the way they are reported.
- What bit strings were encoded for the experimental results? Was the bit string fixed for each image, or was it randomly sampled for each image separately? Are there any bit strings that are harder to encode than others, e.g. all 0 or all 1s? Intuitively, I do not expect it to be the case, but I wonder if the authors carried out any experiments.
- What hardware was used for the encoding is unspecified, which makes the reported decoding times less useful. It would be very good if the authors could specify if the encoding was performed on a GPU and if yes then how long it takes on a CPU and maybe even mobile devices. In a similar vein, the actual number of bits encoded is also unspecified. It would be good if the authors could maybe investigate how long the compression takes per pixel on average with error bars for example.
- As an extension of the second point, what is the biggest message the authors experimented with encoding?
- It is unclear to what the sentence at the top of page 8 is referring. The claim is that FNNS achieves 0% error at 2bpp with a 0.5% detection rate,
  but Table 5 doesn't seem to bear this out for any of the datasets. Is this sentence misplaced perhaps?

## Questions
   - As I understand, the two sources of decoding errors are clipping the difference (line 6 in Alg 1) and clipping the addition of the delta (line 7 in Alg 1).
     Did the authors investigate which of these steps tends to cause more errors? Could the issue be partially resolved by first reparameterizing the image $X \in [0, 1]^{W \times H \times 3}$
     to $ Z \in R^{W \times H \times 3}$ by applying an inverse sigmoid transform $Z = \sigma^{-1}(X)$ and optimizing $Z$ instead of $X$? This would ensure that the second clipping operation never needs to be performed, right?
   - This might be a simple point, but I wonder how robust the encoding is to the random seed used in FNNS-R to sample the NN weights in the following sense.
     Imagine the message is encoded using weights $w_1$ sampled randomly with seed $s_1$. What is the decoding error rate with weights $w_2$ sampled with $s_2$? Assuming that the
     NN predictions are iid Bernoulli $1/2$ variables, the decoding error for a random bit string should be around 50%. Is this indeed a correct intuition?

## After author rebuttal

The authors have addressed my concerns satisfactorily, and I believe that the final version of the paper will be a strong contribution to the literature.


**Summary Of The Paper:**

The paper proposes a novel neural image steganography method. The fundamental idea is that the sensitivity of neural networks to adversarial perturbations can be leveraged to encode an arbitrary hidden message into imperceptible perturbations. The authors' method heavily outperforms current classical and neural image steganography methods. They also perform several relevant experiments, such as robustness of the method to lossy compression and avoiding steganalysis.

The authors also propose to apply their method to protect persons' identities in pictures by detecting all faces present in a picture, in-painting them with a GAN and hiding the picture of the original face in the GAN-generated image.


**Summary Of The Review:**

The core idea presented in the paper is simple and quite general, going beyond image steganography. The authors heavily outperform current steganography algorithms with their method and the community would benefit from its publication.

However, the article has some weaknesses that if the authors can address I will be happy to raise my score.

---

> ### Author Response · Authors · 2021-11-17
> **Response to reviewer 3seg**
>
> Thank you for your detailed review and constructive feedback. Here are some clarifications about our experimental setup:
>
> * We randomly initialized the bit strings; each bit in the string is independently sampled from a Bernoulli-distribution with p=½. We did not observe that any of the randomly initialized strings are harder. It is possible that the all 0s or all 1s string might be harder but that will depend on the initialization of the fixed network. In general we keep that zero-mean and symmetric to not bias the outputs in either direction. We will include experiments on these and more pathological sequences in the final version.
>
> * The encoding is performed on a NVIDIA GTX 1080 GPU and is significantly slower if run on CPU only. However, we have not optimized the code explicitly for CPU or mobile GPU cards, which could be an interesting avenue for future work.
> The images we experimented with have resolution 512 x 512, which results in 1,048,576 hidden bits at 4-BPP.
> Below is a more detailed table that shows the amount of time required for optimizing different numbers of bits with different methods (with standard deviations in parentheses) on a NVIDIA GTX 1080 GPU. We will add this to the camera ready version of the paper as well.
> | Dataset | Method    | 1 bit        | 2 bits         | 3 bits        | 4 bits        |
> | ------- | --------- | ------------ | -------------- | ------------- | ------------- |
> | CelebA  | FNNS - D  | 9.77 (3.46)  | 13.42 (5.17)   | 30.81 (21.20) | 47.39 (15.32) |
> | CelebA  | FNNS - DE | 7.47 (2.20)  | 11.31 (4.90)   | 35.65 (12.32) | 45.39 (16.78) |
> | CelebA  | FNNS      | 45.94 (5.23) | 124.85 (11.29) | 151.94 (7.42) | 152.18 (8.15) |
> | Div2K   | FNNS - D  | 4.95 (1.08)  | 10.53 (9.94)    | 44.39 (5.27)  | 44.29 (5.40)  |
> | Div2K   | FNNS - DE | 4.86 (0.54)  | 8.09 (5.05)    | 43.82 (4.75)  | 44.13 (6.73)  |
> | Div2K   | FNNS      | 42.18 (4.17) | 114.44 (6.48)  | 156.91 (3.88) | 159.19 (4.63) |
> | MS-COCO | FNNS - D  | 7.41 (2.50)  | 10.61 (5.91)    | 37.72 (15.27) | 48.39 (9.40)  |
> | MS-COCO | FNNS - DE | 5.13 (1.97)  | 7.04 (2.34)    | 32.76 (16.20) | 48.29 (9.63)  |
> | MS-COCO | FNNS      | 47.35 (4.23) | 131.85 (12.31) | 182.47 (6.49) | 184.39 (5.74) |
>
> * We experimented with up to 6 bits per pixel on an image of size 512 x 512 (i.e. 1,572,864 total bits). Please refer to Appendix D for our 5-6 BPP results.
> * The sentence at the top of page 8 is a typo. Sorry, we will remove it in the camera ready.
>
> Answers to your questions:
> * Source of errors: The main source of error is clipping the addition of the delta. Your suggestion was interesting and we tried it. However, we see no improvement in performance. This is probably because the final representation space is still limited to  $256^{W \times H \times 3}$ even when we perform the optimization in the reparameterized space.
> * Decoding error with a network that has different weights: Your intuition is absolutely correct. These images are optimized for a very specific network. If you sample new weights for the same network, the decoding error will be 50%.
>
> We hope our responses addressed your concerns. Please let us know if you have any additional questions.

---

> > ### Comment · Reviewer_3seg · 2021-11-18
> > **Response to the authors' comments**
> >
> > I thank the authors for their detailed response.
> >
> > I have a few more questions/suggestions:
> > - I believe that the numbers reported in the above table are in seconds. I wonder then, does the encoding time scale linearly with the image size? Given that the authors only experimented with a fixed image size puts my concern to rest about the meaningfulness of the results, but I would be interested if reporting a metric such as seconds/bit would make sense across different resolutions.
> > - Reviewer Cqai has raised concern in their review about the novelty of the authors' work, citing a couple of works on adversarial methods, which the authors did not address in their response to the reviewer. Could the authors comment on the relevance of these methods and how their work differs from those?
> > - I am somewhat surprised that the parameterization did not help with the error rate, given the authors' comment that the addition of the delta is the main source of the error. If the error rate does not improve, then it appears to me that it is in fact the quantization that is introducing most of the error. In any case, I believe that with a slightly different parameterization, the optimise-project loop in Algorithm 1 could be completely avoided, and could be replaced with a single call to the optimiser. Since the original image $X$ is known before the optimisation, the range of admissible solutions can be easily determined ahead of time pixel-wise as $L = \max(0, X - \epsilon) \leq \tilde{X} \leq \min(1, X + \epsilon) = U $. Then, optimizing $Z$ such that $\tilde{X} = L + (U - L) \sigma(Z)$ should mean that all possible error apart from quantizing the solution is eliminated and there is no need for the loop, correct? Could the authors investigate the effect of quantisation error using this suggestion perhaps?
> >
> > Edit: I think with some additional effort, the quantization step could also be circumvented, thus obtaining a method that should be almost guaranteed to be error-free, so long as the objective can be optimized perfectly. The observation is that, similar to the above suggestion, given an input image and $\epsilon$, the set of admissible pixel values can be easily determined before the optimization. In the paper, the authors report that they set $\epsilon = 0.3$. As I understand this means that the maximum allowed variability in "true pixel space" is $L = \max(0, I -128 * \epsilon) \approx \max(I - 39, 0)  \leq \tilde{I} \leq \min(255, I + 39) \approx \min(I + 128 * \epsilon, 255) = U$,
> > where $I \in [256]^{W \times H \times 3}$ the original image and $\tilde{I}$ is the stego version. This means for each subpixel $\tilde{I_{ijc}}$ we get a certain valid range of pixel values which we could represent as a vector $R_{ijc} = (L_{ijc}, L_{ijc} + 1, ... U_{ijc} - 1, U_{ijc})$. Then, for each subpixel, the problem becomes finding the appropriate value of $R_{ijc}$ to choose for the stego image. To solve this, we can just use the standard softmax relaxation of argmax and use gradient-based optimization. Concretely, For each pixel initialize a vector of $1$s $C_{ijc} = (1, ..., 1) \in \mathbb{R}^{U_{ijc} - L_{ijc}} $. Then, for the temperature variable $t$, taking $R_{ijc}^* = \langle \mathrm{softmax}(C_{ijc}/t), R_{ijc} \rangle$ is a soft approximation to picking one of the elements of $R_{ijc}$, with taking $t \to 0$ giving a hard choice from $R_{ijc}$, where $\langle \cdot, \cdot \rangle$ is the standard dot product. Now, We can form the stego image as $\tilde{I_{ijc}} = R_{ijc}^*$ and take $\tilde{X} = \tilde{I}/255$. Now, optimising $C$ and taking $t \to 0$ slowly during optimization should allow to optimise (almost) directly in pixel space. This should be fairly simple to implement, could the authors try and see if this approach helps at all?

---

> > > ### Author Response · Authors · 2021-11-19
> > > **Response to reviewer 3seg**
> > >
> > > Thank you for following up.
> > >
> > > * The reported times were in seconds. We can definitely report the times in terms of seconds/bit for images of different sizes (this will likely be of interest to many people). The same images for the CelebA dataset are available in different sizes. We will add a table that shows the numbers for the different sizes. The encoding time scales approximately linearly; for every factor of 4 increase in the number of pixels, the encoding time increases by a little less than a factor of 4.
> > > * The papers that reviewer Cqai points out (CNN-Based Adversarial Embedding for Image Steganography and Optimizing Additive Approximations of Non-Additive Distortion Functions) are very different from our contributions. Both the mentioned papers propose using gradients from steganalysis systems with existing traditional steganography methods to evade detection by steganalysis. We instead use imperceptible perturbations directly to perform steganography with a neural decoder. As a consequence, we can hide drastically more bpp in the image (3bpp vs 0.4bpp). The contribution of the mentioned papers is complimentary: similar to this prior work we can also add a loss term in our optimization to evade detection from neural steganalysis. We will clarify the difference between our method and the mentioned ones in the final version.
> > > * Thank you for the well thought out translation to an unconstrained optimization problem. We share your intuitions that unconstrained optimization could be easier than constrained optimization.  Since your initial review, we have implemented and tested variants of your proposals. However, we see no improvements. We believe that the source of error is not the projection step, but rather the restrictions (subpixels remain in [0,1] and change by at most \delta) on the perturbation. Regardless of the parametrization, these restrictions remain and in some cases will not be satisfiable. We will add a section in the appendix to discuss unconstrained optimization.
> > >
> > > Please let us know if there is anything else we can clarify.

---

> > > > ### Comment · Reviewer_3seg · 2021-11-21
> > > > **Response**
> > > >
> > > > I thank the authors again for their response. I believe now that their work would be a strong contribution to the literature and I have raised my score to reflect this.

---

### Decision · Program_Chairs · 2022-01-20

**Decision:**

Accept (Poster)

**Comment:**

This submission proposes a method for steganography, i.e. hiding "secret messages" in images. Specifically, the proposed approach implements a procedure similar to adversarial example generation, where a perturbation is found that a) is imperceptible and b) can be decoded by a fixed decoder. This approach results in the ability to hide a significant amount of information (up to 3 bpp) with essentially no decoding errors. While the resulting perturbations are sometimes detectable by existing methods, the authors nevertheless provide a compelling case that their approach is a significant step forward and demonstrate an interesting new application to face hiding. Reviewers generally found the paper easy to follow and clear and felt the proposed approach provided a new and effective way to perform steganography. However, there were many requests for clarification of the work's situation in the larger literature of steganography and adversarial ML. The authors have clarified their contribution in the rebuttal, and as long as these changes have made it back into the paper, I am happy to recommend acceptance.